# Ceramic-Heterostructure-Based Magnetoelectric Voltage Transformer with an Adjustable Transformation Ratio

**DOI:** 10.3390/ma13183981

**Published:** 2020-09-09

**Authors:** Dmitri Saveliev, Dmitri Chashin, Leonid Fetisov, Mikhail Shamonin, Yuri Fetisov

**Affiliations:** 1Research and Education Center “Magnetoelectric Materials and Devices”, MIREA–Russian Technological University, 119454 Moscow, Russia; chashindv@yandex.ru (D.C.); fetisovl@yandex.ru (L.F.); fetisov@mirea.ru (Y.F.); 2East Bavarian Centre for Intelligent Materials (EBACIM), Ostbayerische Technische Hochschule (OTH) Regensburg, D-93053 Regensburg, Germany; mikhail.chamonine@oth-regensburg.de

**Keywords:** magnetoelectric effect, voltage transformer, composite material, magnetostriction, piezoelectric effect

## Abstract

A voltage transformer employing the magnetoelectric effect in a composite ceramic heterostructure with layers of a magnetostrictive nickel–cobalt ferrite and a piezoelectric lead zirconate–titanate is described. In contrast to electromagnetic and piezoelectric transformers, a unique feature of the presented transformer is the possibility of tuning the voltage transformation ratio *K* using a dc magnetic field. The dependences of the transformer characteristics on the frequency and the amplitude of the input voltage, the strength of the control magnetic field and the load resistance are investigated. The transformer operates in the voltage range between 0 and 112 V, and the voltage transformation ratio *K* is tuned between 0 and 14.1 when the control field *H* changes between 0 and 6.4 kA/m. The power at the transformer output reached 63 mW, and the power conversion efficiency was 34%. The methods for calculation of the frequency response, and the field and load characteristics of the transformer are proposed. The ways to improve performance characteristics of magnetoelectric transformers and their possible application areas are discussed.

## 1. Introduction

Voltage transformers, which are used for converting the amplitude of an alternating voltage and the galvanic isolation of elements of electronic circuits with respect to the dc voltage, are among the most important elements of modern electronics. Currently, compact transformers using the phenomenon of electromagnetic induction [1] and solid-state transformers employing the piezoelectric effect [2] are widely used in low-power electronic circuits. However, both types of transformers have several disadvantages. In particular, electromagnetic transformers contain two voluminous coils, which increase their size and make it difficult to manufacture. Piezoelectric transformers have high input and output impedance. Both types of transformers do not allow one to change the transformation ratio easily and quickly.

It has recently been proposed to employ magnetoelectric (ME) effects in planar composite heterostructures, with layers of a ferromagnetic (FM) and a piezoelectric (PE) materials, to realize various ME devices [3]. ME effects in such structures arise as a result of a combination of magnetostriction in the FM layer and piezoelectricity in the PE layer due to the mechanical coupling between the layers. When an alternating magnetic field *h* is applied to the heterostructure, magnetostriction leads to an alternating deformation of the FM layer, this deformation is transferred to the PE layer and this layer generates an alternating electric voltage *u* (direct ME effect). When an alternating electric field *e* is applied to the PE layer of the structure, it deforms due to the inverse piezoelectric effect, the deformation is transferred to the FM layer. Due to the inverse magnetostriction effect (also known as the Villari effect), a harmonic change in its magnetization *m* (converse ME effect) occurs. The modulation of magnetization is registered using an electromagnetic coil.

In papers [4,5,6,7], step-up ME transformers based on the direct ME effect are described. The transformer [4] was built as a composite structure with layers of an amorphous metal (Metglas) and lead zirconate–titanate (PZT), placed inside an electromagnetic coil. The input voltage *U*_1_, applied to the coil, created an alternating magnetic field and the output voltage *U*_2_ was measured across electrodes of the PZT layer. The voltage transformation ratio *K* = *U*_2_/*U*_1_ at the acoustic resonance frequency of the structure changed by ~2 orders of magnitude in the dc magnetic field *H*. In the transformer [5], a ring of Terfenol and PZT layers was used, on which an input coil was wound. The transformer had a transformation ratio *K* of about 30. The transformer [6] used a slab made of a Terfenol alloy and a longitudinally poled PZT slab, connected in series. The input voltage was applied to a coil wound on a Terfenol slab, and the output voltage was measured from the electrodes of the PZT piece. The voltage transformation ratio reached 130 at the resonance frequency. In [7], a piezoelectric transformer based on a lead magnesium niobate-lead titanate (PMN-PT) crystal placed between two plates of Terfenol with an electromagnetic coil wound on the structure is described.

Hitherto, most of the step-up ME transformers employed FM Terfenol layers. This led to relative high control magnetic fields *H* ≈ 24–64 kA/m. In order to decrease the control power, it is desirable to keep a control magnetic field as low as possible. It will be shown below that, by employing a Ni-Co ferrite as FM layers, it is possible to have a control magnetic field *H* as low as ≈ 5 kA/m. All previously reported realizations of ME step-up transformers [4,5,6,7] used different designs. Hitherto, it is no clear which design of a ME transformer is optimal.

In [8,9], step-down ME transformers based on the converse ME effect are described. The transformer [7] was an electromagnetic coil wound on a three-layer Ni-PZT-Ni structure. The input voltage *U*_1_ was applied to the electrodes of the PZT layer, and the output voltage *U*_2_ was measured on the coil. The transformation ratio *K* was equal to 4.6 × 10^−^^4^ in a broad frequency range. Finally, a transformer was described by Wang et al. [9], which was similar in design to a conventional electromagnetic transformer with two windings, but the cores of the coils were made of multi-layer Metglas-PZT structures. Owing to the converse ME effect, the application of the control voltage to the PZT layers made it possible to alter the magnetic permeability of the cores, which led to a change in the transformation ratio *K* of the device from 0 to 0.2. In the paper by Wang et al. [10], a voltage-controlled static magnetic device called magnetic flux valve employing the converse ME effect was proposed for an adjustable-voltage-ratio transformer, where the transformation ratio *K* varied from 0 to 2.12.

In this work, a step-up ME transformer based on a planar heterostructure is fabricated and studied in detail. Both layers of this transformer, magnetostrictive and piezoelectric, are made of dielectric ceramics in order to reduce eddy-current losses. The key element of the proposed original design is in the usage of dielectric FM layers. This allows one to use the end electrodes (see Figure 1) and therefore take an advantage of the piezoelectric modulus *d*_33_, which is approximately two times larger than the *d*_31_ modulus. Furthermore, we succeed to obtain a high value of the electromechanical quality factor *Q* ~ 10^2^. We believe that all these enhancements enabled the output voltage to reach ~10^2^ V and power ~63 mW, which is significantly larger than in previously reported realizations.

## 2. Experimental Methods 

The composite heterostructure and the design of the ME transformer are schematically shown in Figure 1a,b, respectively. The main element of the transformer is a three-layer heterostructure containing a PE layer sandwiched between two FM layers. The PE layer with dimensions of 20 × 10 mm and a thickness *a*_p_ = 2 mm is made of transformer piezoceramics Pb(Zr,Ti)O_3_ (PZT-47 type, manufactured by the JSC Research Institute “Elpa”, Russia [11]). The piezoceramics has a piezoelectric modulus *d*_33_ = 290 pC/N, high electromechanical quality factor *Q* = 900, an electromechanical coupling coefficient *k*_p_ = 0.56, a dielectric loss tangent tan *δ* < 0.6, and the Curie temperature *T*_C_ = 270 °C [11]. Ag-electrodes were deposited on the end faces of the PE layer and it was poled in the direction of the long axis. The capacitance between the electrodes of the PE cell was *C*_2_ = 66.3 pF. The FM layers were made of a magnetostrictive nickel–cobalt ferrite of the composition Ni_0.99_Co_0.01_Fe_2_O_3_ (manufactured by META, Moscow [12]). Nickel ferrite with a low Co-content was chosen because it has low dielectric losses and a sufficiently high magnetostriction in low magnetic fields. Each layer had dimensions of 20 mm × 10 mm and thickness *a*_m_ = 0.5 mm. The layers had a saturation magnetization *M*_S_ = 0.33 T, a saturation magnetostriction *λ*_S_ = 26 × 10^−6^, initial magnetic permeability *μ* = 51, a magnetomechanical coupling coefficient *k*_m_ = 0.2, and Curie temperature *T*_C_ = 500 °C. The layers of piezoceramics and ferrite were coupled under pressure using a cyanoacrylate adhesive.

The structure was placed inside a 20 mm long electromagnetic coil, containing 120 turns of a wire with a thickness of 0.2 mm. The coil generated an alternating magnetic field *h* with the variable frequency *f*. The structure was rigidly fixed in its central transversal plane for the most efficient excitation of the fundamental mode of longitudinal acoustic vibrations. The resistance and inductance of the coil with the structure inside were *R*_1_ = 2.3 Ω and *L* = 168 μH, respectively. A control magnetic field *H* = 0–15.9 kA/m was applied parallel to the long axis of the structure and the axis of the coil using an electromagnet.

Figure 1c shows a simplified equivalent circuit of the transformer. The input inductive circuit is a coil with an active resistance *R*_1_ and an inductance *L*. The output capacitive circuit is a PE layer of the structure with an active resistance *R*_2_ and a capacitance *C*, connected in parallel. The mechanical part is located between them. The magnetomechanical coupling coefficient for the FM layer is *k*_m_, and the electromechanical coupling coefficient for the PE layer is *k*_p_. The load resistance *R*_L_ is connected in parallel with the output circuit of the transformer. Figure 1d is a picture of the device.

During the measurements, the voltage $U_{1}\cos(2\pi ft)$ from a generator (AKIP 3409/4), with an amplitude *U*_1_ up to 8 V and a variable frequency *f* = 0–200 kHz, was applied to the input coil of the transformer. The output voltage of the transformer *U*_2_ was measured at the load resistance *R*_L_. Both input and output voltages were measured using a voltmeter (AKIP 2401) with an input impedance of more than 10 MΩ. The voltage transformation ratio of the transformer was determined as *K* = *U*_2_/*U*_1_. To measure the input power *P*_1_ of the transformer, a shunt resistor was connected in series with the coil to determine the current *I*_1_. The input active power was calculated by the formula *P*_1_ = (1/2)*U*_1_*I*_1_cos(*φ*), where *φ* is the phase shift between voltage and current. The active power in the output circuit was calculated as $P_{2}=U_{2}^{2}/2R_{L}$. The impedance of the transformer was measured using a panoramic *RLC*-meter (AM-326). The transformer characteristics were recorded for the cases when the frequency *f* and the amplitude *U*_1_ of the input voltage, the control magnetic field *H* and the load resistance *R*_L_ were varied.

## 3. Results and Discussion

In this section, we provide the measured frequency response, field characteristics, and amplitude characteristics. We demonstrate the possibility of tuning the transformation ratio, and describe a method for calculation of main characteristics of the transformer. The ways to improve the performance of ME transformers are discussed in the end of this section.

### 3.1. Frequency Response of the Transformer

Figure 2a shows a typical measured amplitude-frequency response of a transformer with an input voltage *U*_1_ =1 V and a constant magnetic field of *H* = 6.37 kA/m for the open-circuit condition (at *R*_L_ = ∞). One resonance peak around the frequency *f*_0_ ≈ 99.02 kHz was observed in the frequency response. The resonance quality factor was estimated from the width of the resonance curve *δf* at a height of 0.707: *Q* = *f*_0_/δ*f* ≈ 143. The voltage transformation ratio at the resonance frequency is *K* = *U*_2_/*U*_1_ = 14.1. Figure 2b presents the measured dependence of the output impedance magnitude *Z* and its phase *Φ* on the frequency of the input voltage. The minimum impedance value *Z* = *R*_2_ ≈ 17.7 kΩ was observed near the frequency *f*_0_. The voltage phase near the resonance frequency, as can be seen, changes by 180°.

This form of the frequency response of the transformer is due to the characteristics of the resonant ME effect in composite structures. It was shown [13] that the coincidence of the frequency of the excitation magnetic field with the frequency of the acoustic anti-resonance increases the amplitude of the alternating strain in the structure by a factor of *Q*. This leads to an increase in the ME voltage. In this case, there is a longitudinal acoustic resonance in the structure. The resonance frequency can be estimated by the formula for the frequency of the fundamental mode of longitudinal vibrations of a free rod of length L: $f_{0}=\left(1/2L\right)\sqrt{Y/\rho }$, where the effective values of the Young's modulus *Y* and the density ρ are given by the formulas $Y=\left(Y_{m}a_{m}+Y_{p}a_{p}\right)/\left(a_{m}+a_{p}\right)$ and $\rho =\left(\rho_{m}a_{m}+\rho_{p}a_{p}\right)/\left(a_{m}+a_{p}\right)$, respectively. Using the known values of the Young's moduli and the densities of the constitutive layers (*Y*_m_ = 17.9∙10^10^ N/m^2^, *Y*_p_ = 7.7∙10^10^ N/m^2^, ρ_m_ = 5.2∙10^3^ kg/m^3^, ρ_p_ = 7.7∙10^3^ kg/m^3^, the subscripts “m” and “p” correspond to FM layer and PE layer, respectively), we find *f*_0_ ≈ 100.4 kHz. The obtained value of the resonant frequency is in good agreement with the measured one.

### 3.2. Control of the Transformation Ratio

A unique feature of the ME transformer, in comparison with electromagnetic and piezoelectric transformers, is the ability to control the voltage transformation ratio using an external magnetic field. Figure 3 shows the measured dependences of the voltage transformation ratio *K* on the frequency *f* of the input voltage in the absence of a load resistance. The measurements were performed at various values of the control field *H* = 0–15.9 kA/m and the input voltage *U*_1_ = 1 V. It is seen that an increase in the field *H* leads to a strong change in the transformation ratio *K* and a small shift in the resonance frequency *f*_0_.

Figure 4 demonstrates the dependences of the transformation ratio *K*, the resonance frequency *f*_0_, and the quality factor of resonance *Q* on the control magnetic field *H*, derived from the data in Figure 3. It can be seen that with an increasing field, the transformation ratio *K* increases approximately linearly from zero to a maximum value of *K* = 14.1 in the field *H*_m_ ≈ 6.37 kA/m, and then monotonously declines with a further increase in the field. The resonance frequency *f*_0_ grows almost linearly by 0.4% with the increasing magnetic field. The resonance quality factor decreases from *Q* ≈ 200 in the absence of a field to a minimum value of *Q* ≈ 143 in the same field *H*_m_ ≈ 6.37 kA/m, and then rises again to *Q* ≈ 189 at *H* = 15.9 kA/m.

The capability to control the voltage transformation ratio is caused by the dependence of the magnitude of the ME effect in composite structures on a constant field [14,15]. In the work by Dong et al. [16], a formula was obtained for the amplitude of the open-circuit voltage generated by a structure with a longitudinally poled PE layer and a longitudinally magnetized FM layer due to the ME effect. After some algebra, the formula can be rewritten as:(1)$$U_{ME}\left(H\right)=AQd_{33}q_{33}\left(H\right)h,$$where *A* is a constant coefficient depending only on the size and mechanical parameters of the structure layers, *Q* is quality factor of the acoustic resonance, *d*_33_ is the piezoelectric modulus of the PE layer, $q_{33}\left(H\right)={\partial \lambda_{33}/\partial H|}_{H}$ is the piezomagnetic coefficient (derivative of the magnetostriction with respect to the magnetic field) of the FM layer, *λ*_33_(*H*) is the field dependence of the FM layer magnetostriction and *h* is the amplitude of the alternating magnetic field generated by the coil. The index “33” means that the piezoelectric modulus and the piezomagnetic coefficient are measured in the direction of the electric field *E* and magnetic field *H*, respectively. Further, for simplicity, we omit this index. The dependence *q*(*H*) in Equation (1) leads to the dependence *U*_ME_(*H*).

To confirm this conclusion, the field dependence of magnetostriction λ(*H*) was measured by a strain gauge glued to the surface of the ferrite layer. Then, using the numerical differentiation, the field dependence of the piezomagnetic coefficient *q*(*H*) was found. The obtained dependence *q*(*H*) is shown in Figure 4a. For convenience of comparison, the scale along the vertical axis for *q* is chosen so, that the maxima of the dependences *K*(*H*) and *q*(*H*) visually coincide. It can be seen that the piezomagnetic coefficient *q* initially linearly increases with increasing field *H*, reaches a maximum at the same characteristic field *H*_m_ ≈ 6.37 kA/m, and then decreases as the ferrite layer is saturated. The shapes of the field dependences *K*(*H*) and *q*(*H*) agree qualitatively well. The discrepancy between the curves in the region of large fields can be due to the influence of the Poisson’s effect and the inhomogeneity of the magnetic field inside the FM plates due to the demagnetizing fields, which were not taken into account in the calculations. The dependence of the resonance frequency *f*_0_ and the quality factor *Q* of the structure on the field *H* (Figure 4b) is caused by the dependence of the Young's modulus and the mechanical losses of the ferrite layer on the magnetic field *H*, as observed previously [15].

### 3.3. Load Characteristics

Next, the characteristics of the transformer were measured in dependence on the load resistance. Figure 5 shows the dependence of the transformation ratio *K* on the frequency of the input voltage for different load resistances *R*_L_ = 0–220 kΩ. The measurements were carried out in the optimal bias field *H*_m_ = 6.37 kA/m and an input voltage amplitude of *U*_1_ = 1 V. It can be seen that the transformation ratio is highly dependent on the load, while the resonance frequency and the quality factor of the resonance change slightly.

Figure 6 presents the dependences of the transformation ratio *K*, output power *P*_2_, frequency *f*_0_, and *Q*-factor of resonance on the load resistance *R*_L_ constructed from the data in Figure 5. It can be seen that the transformation ratio increases monotonically from zero to 14.1 with the load resistance increasing up to *R*_L_ = 220 kΩ. In this case, the output power *P*_2_ varies non-monotonically: first it increases from zero and reaches a maximum *P*_m_ ≈ 1.18 mW with a load resistance of *R*_m_ ≈ 18–20 kΩ, and then monotonously decreases to *P*_2_ ≈ 0.9 mW with a further increase in resistance to *R*_L_ = 220 kΩ. A similar type of dependence was observed earlier in [17]. In this case, the resonance frequency *f*_0_ monotonously increases by less than 0.5%, from 98.54 to 99.02 kHz with the increasing load. The resonance quality factor *Q* first steeply decreases with the increasing load resistance from *Q* ≈ 100 to *Q* ≈ 80 at *R*_L_ ≈ 5 kΩ, and then again increases to *Q* ≈ 142 at high load resistances.

To explain the dependences presented in Figure 6a, we use a simplified equivalent circuit for the capacitive part of the transformer as shown in Figure 7. The output circuit is an ac voltage source with an open-circuit voltage amplitude *U*_ME_ and the internal impedance represented by a lossy capacitor (*C*, *R*_2_). The output circuit is loaded by resistance *R*_L_.

The amplitude of the voltage *U*_2_ on the load resistance is given by the formula
(2)$$U_{2}=U_{ME}R_{L}\sqrt{\frac{1+{\left(\omega CR_{2}\right)}^{2}}{{\left(R_{2}+R_{L}\right)}^{2}+{\left(\omega CR_{2}R_{L}\right)}^{2}}},$$
where ω=2πf. For the open circuit condition ($R_{L}\to \infty$) one obtains from Equation (2) that *U*_2_ = *U*_ME_. Thus, the *U*_ME_ is the amplitude of the ME voltage generated by the PZT layer of the structure as a result of the ME effect in the absence of a load (see Equation (1)).

At the resonance frequency *f*_0_ = 99.5 kHz, the resistance and the capacitance of the PZT layer were *R*_2_ ≈ 17.7 kΩ and *С* ≈ 66.3 pF, respectively. The dependence $K\left(R_{L}\right)=U_{2}\left(R_{L}\right)/U_{1}$ calculated using Equation (2) for the amplitude of the input voltage *U*_1_ = 1 V and the amplitude of the ME voltage *U*_ME_ = 15 V is shown in Figure 6a with a dashed line. It can be seen that the theory describes the dependence of the transformation ratio on the load resistance qualitatively well.

For the output power *P*_2_, using Equation (2), we obtain the expression:(3)$$P_{2}=\frac{U_{2}^{2}}{2R_{L}}=\frac{U_{ME}^{2}}{2}\frac{R_{L}[1+\left(\omega CR_{2}{)}^{2}\right]}{{\left(R_{2}+R_{L}\right)}^{2}+{\left(\omega CR_{2}R_{L}\right)}^{2}},$$

This function, i.e., power at the transformer output, reaches its maximum value at *R*_L_ ≈ *R*_2_ ≈ 17.7 kΩ. The dependence *P*_2_(*R*_L_) calculated using Equation (3) with the parameter values *U*_ME_ = 15 V and *R*_2_ = 17.7 kΩ is also shown in Figure 6a with a dashed line. Again, the theory qualitatively well describes the dependence of the output power on the load resistance. To match the calculation results with measurements better, it is necessary to use a full equivalent circuit that takes into account the influence of the mechanical part of the transformer (see Figure 1c) on its output characteristics.

### 3.4. Amplitude Characteristics

Figure 8 presents the dependences of the output voltage of transformer *U*_2_ on the frequency of the input voltage at different amplitudes *U*_1_ under open-circuit condition with *H*_m_ = 6.37 kA/m. It can be seen that the output voltage *U*_2_ increases with increasing input voltage *U*_1_, while the line shape is distorted, and the center frequency *f*_0_ is slightly shifted towards lower frequencies. The relative shift of the resonant frequency did not exceed 1%. A downward frequency shift with increasing *U*_1_ was previously observed in ME resonators. It is caused by a change in the rigidity of the FM layer of the structure under an alternating magnetic field [18].

Figure 9 shows the dependences of the amplitude of the output voltage *U*_2_ at the resonance frequency on the input voltage *U*_1_ for various load resistances *R*_L_, constructed using the data similar to those in Figure 8. The dependences *U*_2_(*U*_1_) are linear with high accuracy in the input voltage range up to 8 V. The slope of the lines, i.e., the voltage transformation ratio *K* = *U*_2_/*U*_1_, depends on the load value *R*_L_, in accordance with the data in Figure 6a. Similar linear dependences, but with smaller transmission coefficients, were observed under different external magnetic fields.

Using the data in Figure 9, we estimated the maximum active power at the output of the transformer *P*_2_ and the maximum power transfer efficiency *η* = *P*_2_/*P*_1_. The power in the output circuit was maximum at the resonant frequency with the optimal load resistance *R*_L_ = 20 kΩ and a control magnetic field of *H*_m_ = 6.37 kA/m (see Figure 6a). At an output voltage of *U*_2_ = 50 V, it was 63 mW. This corresponds to the power density in the heterostructure of approximately 100 mW/cm^3^.

The maximal power transfer coefficient was measured at an input voltage of *U*_1_ = 1 V, load resistance *R*_L_ = 20 kΩ, and control field *H*_m_ = 6.37 kA/m. The current amplitude in the input circuit was *I*_1_ = 7.59 mA, and the phase shift between current and voltage was *φ* = 24^0^. Then $P_{1}=\left(1/2\right)U_{1}I_{1}\cos\phi$ ≈ 3.5 mW, power in the output circuit $P_{2}=U_{2}^{2}/2R_{L}$ ≈ 1.18 mW. The maximum power transfer coefficient of the transformer reached *η* = (*P*_2_/*P*_1_)100% ≈ 34 %. When the input voltage frequency was far from the resonance frequency, the power transfer coefficient decreased to a part of percent. The measurements showed that the power transfer coefficient of the transformer at resonance depended weakly on the amplitude of the input voltage in the investigated voltage range 0 < *U*_1_ < 8 V.

In the presented experimental setup, the control magnetic field *H* was created by oversized electromagnetic coils. The electrical power consumption required to maintain the field *H* had the order of magnitude of ~10^2^ mW. The experiments revealed that the optimum control field *H*_m_ is rather low (≈6 kA/m or ≈7.5 mT in the air). Such fields can be easily created by permanent magnets. The magnetic field can be tuned by changing the arrangement of magnets mechanically. In this case, no power is required to maintain the control field. If ME transformers of larger size will be employed, the power consumption by the control electromagnetic coils may become negligible.

### 3.5. Ways to Improve Characteristics of the Magnetoelectric Transformers

The above results demonstrate the possibility of creating transformers with a controlled voltage transformation ratio based on the ME effect. The output voltage of such transformers can reach hundreds of Volts and the transmitted power can reach hundreds of mW, which will enable their usage in low-power electronic circuits.

There are additional opportunities to improve the performance characteristics of ME transformers. The maximum control magnetic field in the described transformer with an FM nickel–cobalt ferrite layer was *H*_m_ ≈ 6.4 kA/m. This field can be further reduced, if an FM layer made of an amorphous magnetostrictive alloy is used. For an amorphous alloy, for example, FeBSiC (Metglas 2603S3A), the maximum of the piezomagnetic coefficient is achieved in fields *H*_m_ ~0.2–0.4 kA/m [19,20]. This would significantly reduce the size and the electrical power consumption of the control magnetic system of the transformer.

The value of the voltage transformation ratio in the described transformer reached *K* = 14.1. This transformation ratio depends on the ME conversion coefficient and the acoustic quality factor of the composite heterostructure [4,21]. The typical quality factor for heterostructures with PE ceramic layers is *Q* ~10^2^. The quality factor of the structure can be increased up to *Q* ~10^3^–10^4^ if, instead of ceramics, a single-crystal PE layer is employed, for example, of aluminum nitride [22], langatate [23] or piezoelectric semiconductor [24,25]. This makes it possible to increase the transformation ratio of ME transformers up to *K* ~10^3^. Besides, in structures with single-crystal PE layers and thin FM layers, the shift of the resonance frequency with a change in the control field or amplitude of the alternating field is ~2-orders of magnitude less than in structures with ceramic layers [24]. The use of PE layers of single-crystals will stabilize the operating frequency of ME transformers.

For ME gyrators that have a design similar to ME transformers, it was shown that, by optimizing the size and parameters of the composite structure, it is possible to increase the power transfer efficiency up to *η* ≈ 90% [26,27,28,29]. One can hope that similar increase in the power transfer efficiency can be realized for the ME transformers as well.

It is possible to get rid of control magnetic system in ME transformers at all, by using so called “self-biasing” [30]. The self-biasing arises in FM layers made of material with distinct hysteresis or in layers comprising two magnetic materials with different magnetization, and provides effective operation of ME devices at zero external magnetic field.

Finally, it is possible to simplify design of the ME transformer and to reduce its dimensions if not a voluminous coil, but a linear electrical current generated by the input voltage is employed to excite the ME effect in a heterostructure. This current can be driven through a microstrip located near the surface of the structure, or directly through the FM layer of the structure [31,32]. Such a design will allow one to employ well-developed methods of planar technology for the manufacturing of ME transformers.

Thus, the above research results demonstrate for the first time the feasibility of a voltage transformer using the ME effect in a ceramic, laminated ferromagnetic-piezoelectric composite. It is shown that the ME step-up transformer can operate in the range of output voltages up to hundreds of volts and transmit power in the ten to hundred mW range. A unique property of an ME transformer is the tuning of its transformation ratio using a weak magnetic field. ME magnetic-field-controlled transformers can find applications in measuring systems, in low-power automatic control and monitoring systems as well as as an alternative for on-load tap-changing (OLTC) electromagnetic transformers, step-voltage regulators and solid-state transformers for smart grids [33]. However, additional research is required to improve characteristics of the ME transformer, in particular, to increase the transformation ratio, the transformer efficiency, the transformer power rating and to design compact electromagnetic systems for tuning of the control magnetic field.

## 4. Conclusions

It is shown that a ME voltage transformer based on a composite ceramic heterostructure with layers of a nickel–cobalt ferrite and a lead zirconate–titanate allowed one to reach promising performance characteristics with regard to the adjustable voltage transformation ratio, when compared to previous designs. The originality of the proposed transformer is the usage of a dielectric FM ceramics (a Ni-Co ferrite) and positioning of electrodes at the ends of a PE layer, exploiting a larger PE modulus. The transformer operated in the output voltage range from 0 to 112 V, the voltage transformation ratio reached *K* = 14.1. With an optimal load resistance of approximately 20 kΩ, the maximum power at the transformer output reached 63 mW, and the maximum power transfer efficiency reached 34 %. The maximum control magnetic field was relatively low, *H* ≈ 6.37 kA/m. The methods for calculating the characteristics of the transformer showed good agreement with experimental results. Current research is on the way to increase the size of ME transformers and/or to incorporate permanent magnets into an electromechanical field-control system.

## Figures and Tables

**Figure 1 materials-13-03981-f001:**
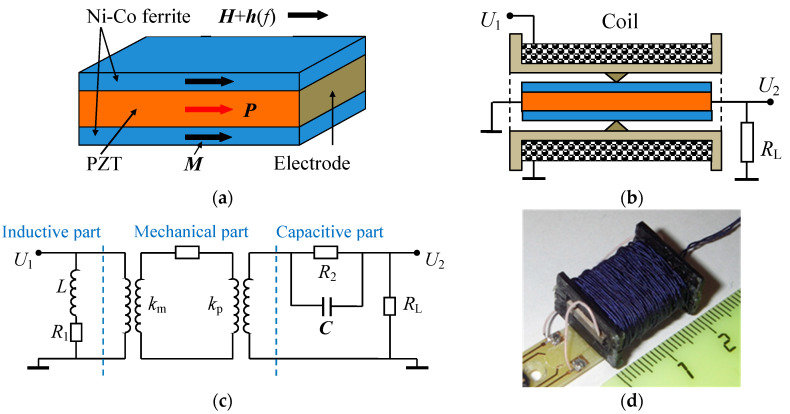
Schematic view of (**a**) the Ni-Co-ferrite-PZT heterostructure and (**b**) the transformer. The arrows denote directions of magnetic field *H*, magnetization *M*, and polarization *P*. (**c**) Simplified equivalent circuit of the transformer. (**d**) Picture of the ME transformer.

**Figure 2 materials-13-03981-f002:**
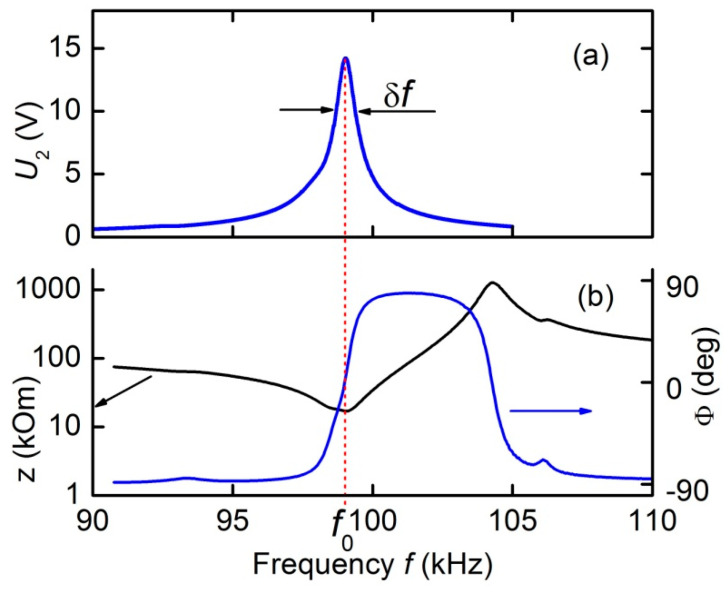
Dependences of the output voltage *U*_2_ (**a**) as well as the magnitude *Z* and the phase *Ф* (**b**) of the output impedance of the magnetoelectric (ME) transformer on the voltage frequency *f*. Measurements were carried out under open-circuit condition.

**Figure 3 materials-13-03981-f003:**
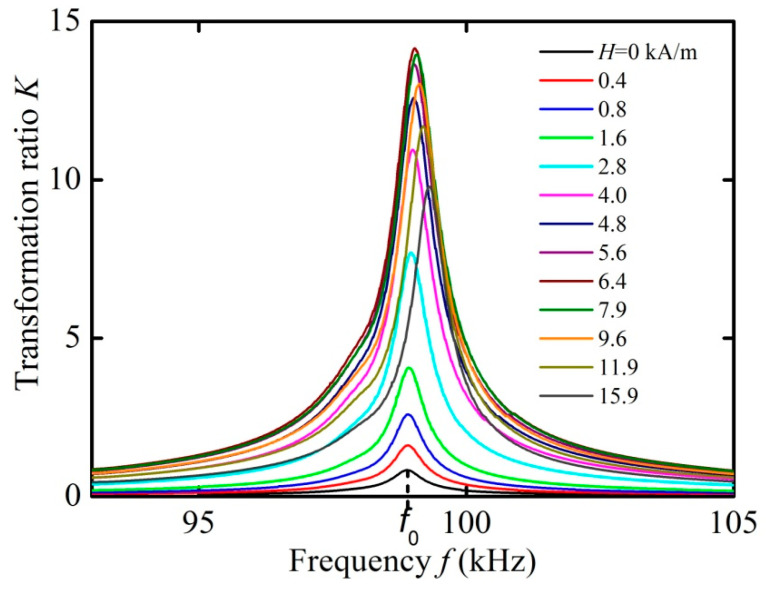
Dependences of the voltage transformation ratio *K* on the frequency *f* for different control fields *H* (0–15.9 kA/m) under open-circuit condition.

**Figure 4 materials-13-03981-f004:**
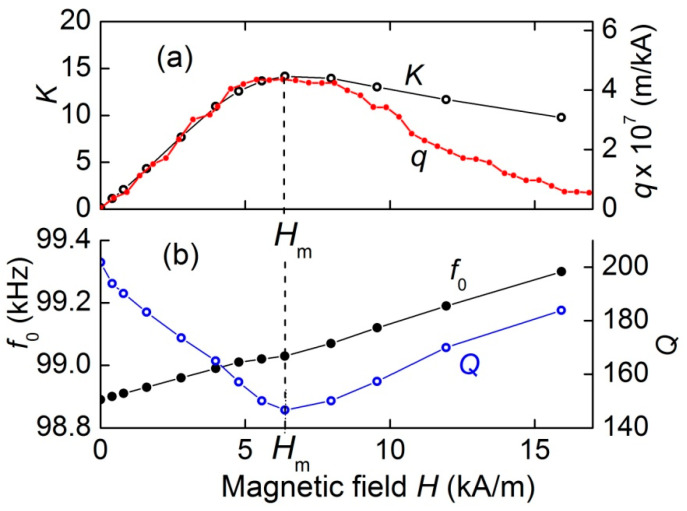
Field dependences of (**a**) the transformation ratio *K* and the piezomagnetic coefficient *q,* and (**b**) the resonance frequency *f*_0_ as well as the quality factor *Q* under open-circuit condition. *H*_m_ is the field of the maximum *K* value.

**Figure 5 materials-13-03981-f005:**
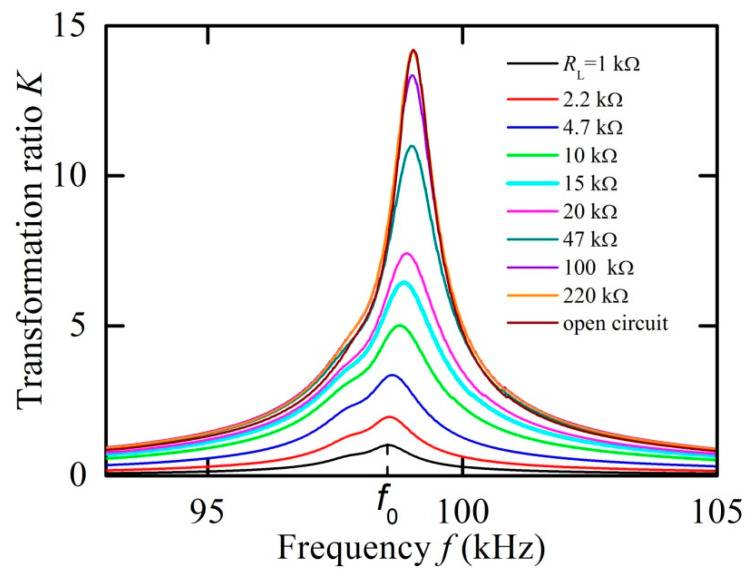
Frequency dependences of the transformation ratio *K* for different load resistances *R*_L_ (1–220 kΩ) at *H*_m_ = 6.37 kA/m, which has been determined under open-circuit condition.

**Figure 6 materials-13-03981-f006:**
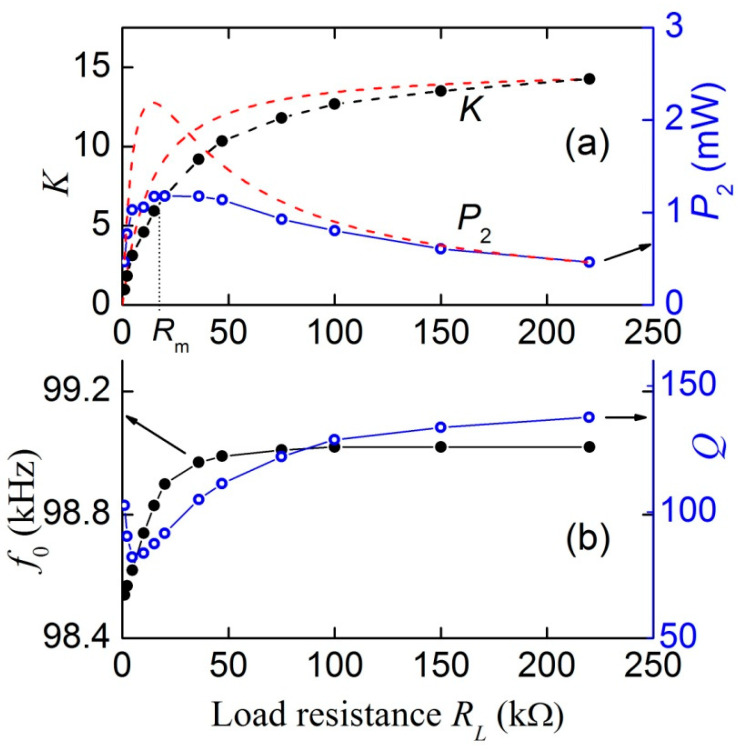
Dependences of (**a**) the voltage transformation ratio *K* and the output power *P*_2_ on the load resistance *R*_L_ and (**b**) the resonance frequency *f*_0_ and the quality factor *Q* on the load resistance *R*_L_ at *H* = 6.37 kA/m. The dashed lines in (**a**) are the calculated dependences. Solid lines are drawn to guide the eye. *R*_m_ is the load resistance for the maximum output power.

**Figure 7 materials-13-03981-f007:**
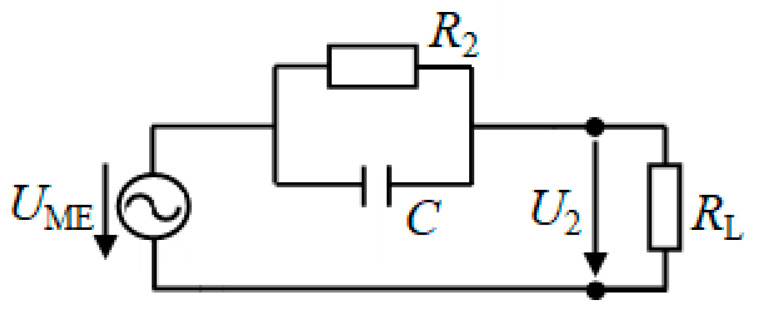
Simplified equivalent circuit of the output capacitive part of the ME transformer. *U*_ME_ is the open-circuit magnetoelectric voltage; *R*_2_ is the output resistance of the transformer, *C* is the output capacitance of the transformer, both connected in parallel; *R*_L_ is the load resistance; *U*_2_ is the measured voltage at the load resistor *R*_L_.

**Figure 8 materials-13-03981-f008:**
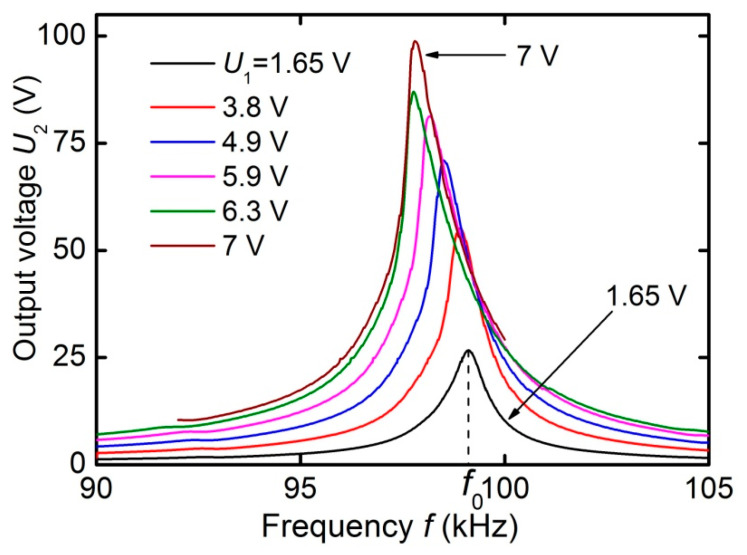
Dependences of the output voltage *U*_2_ of the transformer on the frequency *f* of the input voltage at various amplitudes of *U*_1_ (1.65 V–7 V) under open-circuit condition and *H*_m_ ≈ 6.37 kA/m.

**Figure 9 materials-13-03981-f009:**
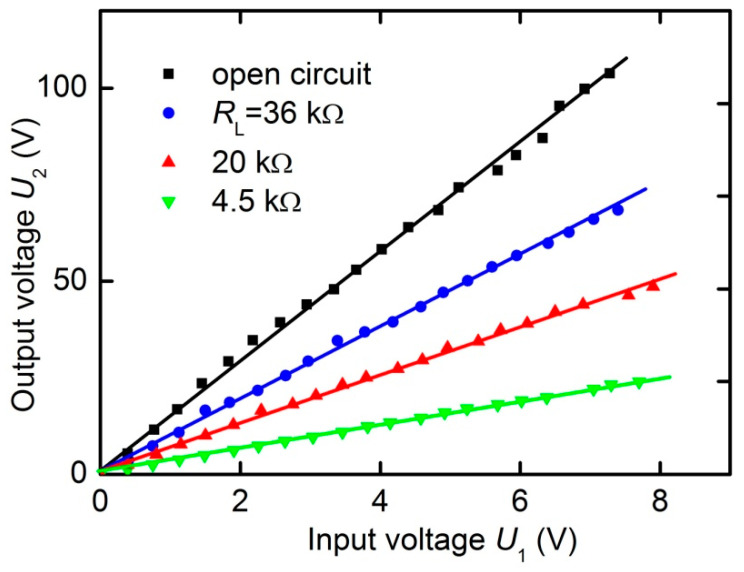
Dependences of the output voltage *U*_2_ on the input voltage *U*_1_ at the resonance frequency *f*_0_ and *H*_m_ = 6.37 kA/m for different load resistances *R*_L_ (4.5, 20 and 36 kΩ) and under open-circuit conditions. Solid lines are the linear approximations.

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
