# Peer review of "Ceramic-Heterostructure-Based Magnetoelectric Voltage Transformer with an Adjustable Transformation Ratio"

_materials, 2020, doi:10.3390/ma13183981_

Round 1

Reviewer 1 Report

The paper is describing step-up voltage transformer utilizing the magnetoelectric effect in a composite heterostructure . Even though neither concept nor approach are original the composite material selection toward minimum eddy current  losses is well justified .  The authors present  strong technical verification experiment to demonstrate reasonably high voltage transformation ratio that is  tuned by the applied magnetic field. This field  is proposed to be creates either by electromagnet or by means of permanent magnets.  It assumed that I no energy is required  to  change magnetic field when permanent magnets are used, however it is still not cleat how it is possible to adjust/move /rotate  that magnets in order to vary the field in vicinity of the transformer and how much power that would require to control the transformer of that design.  While experimental and characterization procedure of the magnetoelectric composite is well described , the materials components ( PZT and Ni0.99Co0.01Fe2O3.) and preparation is still missing some specific details. It is not not enough information on why exactly this NCFO composition was selected and how it was fabricated.  

Author Response

The paper is describing step-up voltage transformer utilizing the magnetoelectric effect in a composite heterostructure. Even though neither concept nor approach are original the composite material selection toward minimum eddy current losses is well justified. The authors present strong technical verification experiment to demonstrate reasonably high voltage transformation ratio that is tuned by the applied magnetic field. This field is proposed to be creates either by electromagnet or by means of permanent magnets. 

  1. It assumed that no energy is required to change magnetic field when permanent magnets are used, however it is still not clear how it is possible to adjust/move /rotate  that magnets in order to vary the field in vicinity of the transformer and how much power that would require to control the transformer of that design. 

Comments: In the described experiments, an electromagnet was used to create a constant field H. The maximum field required for tuning the transformer is small and amounts to ~ 6 kA / m, and the power spent on its creation is about 102 mW. Such a field can be easily created with a small electromagnetic system or with permanent magnets. The development of a control magnetic system for a transformer is a separate task and was not included in this work. Note that with an increase in the size of the transformer, the relative power spent on its tuning will decrease. The corresponding paragraph was added to the text of the article, lines 291-297.

  1. While experimental and characterization procedure of the magnetoelectric composite is well described, the materials components ( PZT and Ni0.99Co0.01Fe2O3.) and preparation is still missing some specific details. It is not enough information on why exactly this NZFO composition was selected and how it was fabricated. 

Comments: We used commercially available materials to make the transformer. Piezoceramics were manufactured by ELPA and magnetostrictive ferrite was manufactured by META. Links to firms' websites are given in the references [11] and [12]. Nickel ferrite with low content of Co was chosen because it has a sufficiently large magnetostriction in a low magnetic fields among commercially available ferrite and low dielectric loss. A relevant sentence has been added to the text, lines 98-100.

Reviewer 2 Report

"Ceramic Heterostructure Based Magnetoelectric Voltage Transformer with Adjustable Transformation Ratio"

The Authors present and discuss results on a voltage transformer employing the magnetoelectric effect in a composite ceramic heterostructure with layers of a magnetostrictive nickel-cobalt ferrite and a piezoelectric lead 14 zirconate-titanate. 

The manuscript is well written, the data is well presented and discussed. However, it needs some revision; it should not be accepted for publication in this present form.

  • Title: meaningful
  • Abstract: meaningful
  • Keywords: meaningful

  1. Introduction
  • this section is comprehensive yet the Authors use long phrases, unnecessary examples, repetitive phrases, and in some cases it's very difficult to follow; please rephrase it
  • the Authors must further insist on the importance and novelty of their work with respect to literature; further explain on your choice on this approach; avoid using superfluous text
  • rephrase the last paragraph of this section to better emphasize your work presented in this manuscript (it's not necessary to present each of the sections here)

  1. Experimental methods
  • section "2. Transformer Design and Methods for Characteristics Measurements" needs to be renamed as such (recommended)
  • provide further and adequate references to the design, the fabrication / synthesis route (if available);
  • further explain on your approach, further present and discuss on the parameters ...

  1. Results and discussion
  • the first paragraph needs to be some sort of introduction of the section: short, concise, explaining the main ideas of this work (a few phrases suffice)
  • section "4. Ways to Improve Characteristics of the Magnetoelectric Transformers" needs to be a part of the "Results and discussion" (as a subsection)
  • further discuss all figures, tables, schematics and graphs in text; provide more meaningful figure captions
  • to conclude, the main comment of this section is that the Authors need to further discuss the results in a more correlated manner; further insist on the importance and novelty of this work with respect to literature; avoid using superfluous text
  • the last paragraph needs to be some sort of a conclusion (briefly presenting the main conclusion of this work)

  1. Conclusion
  • this section is meaningful, but too similar to the abstract; please rephrase it to insist more on the novelty and importance of your work, provide more on the actual and significant values / data

Minor aspects

The Authors need to avoid providing redundant data, superfluous text, or the repetitive use of speculative words or phrases; check the manuscript for typographical errors (language, spelling, punctuation, numbering of sections, etc).

To conclude, the manuscript should not be accepted for publication in its present form. Further revision is needed.

Author Response

The Authors present and discuss results on a voltage transformer employing the magnetoelectric effect in a composite ceramic heterostructure with layers of a magnetostrictive nickel-cobalt ferrite and a piezoelectric lead zirconate-titanate. 
The manuscript is well written, the data is well presented and discussed. However, it needs some revision; it should not be accepted for publication in this present form.

  1. This section is comprehensive yet the Authors use long phrases, unnecessary examples, repetitive phrases, and in some cases it's very difficult to follow; please rephrase it.

Comments: The repetitive phrases are deleted from Introduction. It contains only short descriptions of the principle of operation of known transformers and links only to the papers closest to the topic of our work. This is necessary to explain the novelty of the work.

  1. The Authors must further insist on the importance and novelty of their work with respect to literature; further explain on your choice on this approach; avoid using superfluous text

Comment: At the end of the Introduction, a paragraph (lines 79-86) is added explaining the novelty of this work in comparison with the known ones. 

  1. Rephrase the last paragraph of this section to better emphasize your work presented in this manuscript (it's not necessary to present each of the sections here)

Comment: A paragraph has been added at the end of the introduction. The last paragraph listing the sections of this article has been removed from the text.

  1. Experimental methods Section "2. Transformer Design and Methods for Characteristics Measurements" needs to be renamed as such (recommended)

Comment: The section is renamed to  “Exprerimental methods”.

  1. Provide further and adequate references to the design, the fabrication / synthesis route (if available).

Comments: The design of the proposed transformer is original and therefore we cannot give a reference. To fabricate the transformer we used commercially available piezoelectric and magnetostrictive materials supplied by companies ELPA and META. Corresponding references [11] and [12] are inserted in the reference list. 

  1. Results and discussion. The first paragraph needs to be some sort of introduction of the section: short, concise, explaining the main ideas of this work (a few phrases suffice)

Comment: Introducing first paragraph is inserted in the section “Results and discussion”

  1. Section "4. Ways to Improve Characteristics of the Magnetoelectric Transformers" needs to be a part of the "Results and discussion" (as a subsection).

Comment: Section 4 “Ways to Improve Characteristics of the Magnetoelectric Transformers” is formed as a subsection 3.5 of the section 3 “Results and discussion.”

  1. Further discuss all figures, tables, schematics and graphs in text; provide more meaningful figure captions

Comment: Captions for figures 2-9 are improved.

  1. The last paragraph needs to be some sort of a conclusion (briefly presenting the main conclusion of this work)

The concluding paragraph is added in the end of Section 3, which summarize main results of the work and indicate possible areas of the ME voltage transformer applications, lines 336-346.

  1. This section is meaningful, but too similar to the abstract; please rephrase it to insist more on the novelty and importance of your work, provide more on the actual and significant values / data

The conclusion is rewritten in order to stress the novelty of the work and provide significant results obtained. 

  1. Minor aspects: The Authors need to avoid providing redundant data, superfluous text, or the repetitive use of speculative words or phrases; check the manuscript for typographical errors (language, spelling, punctuation, numbering of sections, etc).

Several misprints and errors are corrected in the text of the submission.
